# Family Resilience Progress from the Perspective of Parents of Adolescents with Depression: An Interpretative Phenomenological Analysis

**DOI:** 10.3390/ijerph20032564

**Published:** 2023-01-31

**Authors:** Yinying Zhang, Chongmei Huang, Min Yang

**Affiliations:** 1Xiang Ya Nursing School, Central South University, Changsha 410017, China; 2School of Nursing, Tongji Medical College, Huazhong University of Science and Technology, Wuhan 430074, China

**Keywords:** family resilience, adolescents with depression, parents, interpretative phenomenological analysis, processes

## Abstract

Family resilience plays an important role in the healthy family development of adolescents with depression, but few studies have explored the specific process of family resilience. This study aims to explore the dynamic processes of family resilience from parents of adolescents with depression. Data were collected from 14 Chinese parents of adolescents with depression by interpretative phenomenological analysis method. Four themes and 12 sub-themes emerged: (1) decompensation phase: (i) misinterpretations of illness, (ii) heavy psychological burden, (iii) chaotic rhythms in family; (2) launch phase: (i) potential influences of labeling, (ii) we must cure my child anyway, (iii) begin adjusting to family roles; (3) recovery phase: (i) family reflection, (ii) subsequent reorganization of family resources, (iii) ultimately establishing a new balance; (4) normality phase: (i) adaption for medical seeking process, (ii) actively lower expectations, (iii) concerns of future needs. Mental health professionals could provide targeted suggestions to help the parents achieve family resilience by assessing its different phases.

## 1. Introduction

Depression affects 20% of the world’s population, and its typical age of onset is adolescence [1]. Adolescent depression is a chronic disease that is prone to relapse and disability [2]. Compared with adults with depression, adolescents with depression have a higher risk for future negative outcomes, such as a greater likelihood of attempted suicide and other psychiatric co-morbidity [3]. Recent epidemiological studies suggested that the worldwide prevalence of adolescent depression was increasing significantly, especially among female adolescents [4,5]. The worldwide prevalence of major depressive disorder in adolescents has been reported to be 1.3% [6]. In China, the pooled point prevalence of the major depressive disorder in adolescents is similar to worldwide figures. Based on the point prevalence of 1.3%, there are about 2.3 million adolescents with major depressive disorder in China [7].

The family system, especially the parents, plays a vital role in the rehabilitation of adolescents with depression [8]. Family environment is an important moderator in the relationship between depression treatment and adolescents’ suicidal ideations [9]. Spending time with parents can effectively reduce the symptoms of adolescents with depression [10]. However, meeting the long-term care demands of adolescents with depression often brings too many challenges for the family system, such as the financial burden of treatment costs [11], more family arguments [12], and social stigma [13]. The recovery of adolescents with depression will be affected if those challenges are not well coped with. Fortunately, there is evidence that some families of children with mental illness demonstrated satisfactory coping skills to better face these challenges [14] and used their experiences to promote healthy family development [15]. 

Family resilience is an important indicator of family systems [16]. Family resilience deserves attention for the reason that it regards the family as a unit composed of the interaction of each individual resilience, and its effect is higher than the individual resilience [17]. When the family system is challenged by illness, family resilience not only produces a direct effect on the improvement of the patient’s quality of life [18] but also improves the sleep quality and depressive symptoms of their family members [19], thereby reducing their caregiving burden [20]. Furthermore, family resilience has been identified as an effective family-centered intervention to promote family communication so as to improve the suffering of parents and children, and these improvements are sustained even after the intervention is discontinued [21]. 

Different authors have different understandings of family resilience. McCubbin, H. I., and McCubbin, M. A. [22] describe resilience as a family characteristic or pattern under stressful or adverse circumstances. However, the crisis faced by families is dynamic in nature; therefore, it is necessary to consider the dynamic process of a family coping with illness when studying family resilience [23]. Walsh [16] emphasizes an ecological and developmental view in understanding family resilience and defines it as the function of the family system in coping with adversity. Key progress in Walsh’s family resilience framework consists of three main components: (1) belief systems, including making meaning of adversity, positive outlook, transcendence, and spirituality; (2) organizational patterns, including flexibility, connectedness, social and economic resources; (3) communication/problem-solving, including clarity, open emotional expression, and collaborative problem-solving [16]. However, this framework does not explore in depth the specific performance of family resilience processes in the different phases of illness [24]. In order to facilitate a more comprehensive and nuanced perspective for exploring the family resilience process of parents of adolescents with depression and its specific manifestations at different phases of illness, this study divides the phase of illness of adolescent depression into four phases: newly diagnosed phase, acute treatment phase, rehabilitation phase, and recurrence phase. 

Although family resilience focuses on its influencing factors on family resilience and the dynamic processes that promote the positive adaptation of the family unit and all members [16], there is limited research on the dynamic processes of family resilience. At present, the research on family resilience of children and adolescents with chronic diseases is more quantitative research to investigate the level and impact factors of family resilience [15,19,25]. Some qualitative studies have explored family resilience in adult patients with mental illness, but these studies have also focused more on risk and protective factors associated with family resilience [26,27]. A recent qualitative study explored the experiences of family resilience from the view of the adult children of parents with bipolar disorder, but it did not explore the specific performance of family resilience processes during the different phases of illness [28]. Up to now, few studies have explored the dynamic process of family resilience during the different phases of illness among patients with mental disorders, and even fewer in exploring the family resilience of adolescents with mental disorders. 

Since the successful implementation of family resilience mainly depends on the active engagement of parents of adolescents with depression, this study used a qualitative approach to explore the dynamic processes of family resilience in each illness phase among adolescents with depression from the perspectives of parents. It is intended to lay the groundwork for future research to develop a theoretical framework or research instrument on family resilience in parents of adolescents with depression. 

## 2. Materials and Methods

### 2.1. Design

This qualitative study was the second part of a larger explanatory mixed-methods project that quantitatively investigated family resilience and its influencing factors among parents of adolescents with depression. It also explored their experiences with family resilience. After the quantitative cross-sectional survey, we calculated scores of family resilience among the parents and used this as a sampling strategy for the selection of qualitative participants. Some general information that has been shown to influence family resilience in the quantitative part, including relationships with adolescents with depression (father or mother), employment, educational level, and marital status, was also considered as a sampling strategy for the qualitative part. This study used an interpretative phenomenological analysis (IPA) design [29]. IPA recognized that the research process was dynamic and acknowledged the “double hermeneutic“, whereby the participant seeks to make sense of their world, and the researcher seeks to make sense of the participant’s sense-making [29]. IPA was chosen as it was a flexible and in-depth approach that explored in detail how meanings were constructed by participants within both a social and a personal world according to their particular experiences [29]. Since family resilience was a concept with positive meaning, IPA had proven to be an effective approach to understanding family members’ perspectives on the construction of meanings of family resilience in taking care of adults with mental disorders [27,28]. Hence, it was appropriate for this study to use IPA to explore their understanding and construction of meanings on family resilience from the perspective of the parents of adolescents with depression.

### 2.2. Data Collection

This study was conducted at the inpatient department of children and adolescent psychiatry of a National Clinical Mental Health Center of a tertiary hospital in Hunan province, China, from December 2020 to January 2021. The center was equipped with 45 beds in the department of children and adolescent psychiatry and provided services for approximately 1200 inpatients per annum from all over the country. 

Participants for this study were recruited by purposive sampling technique with maximum variation sampling strategy, which contributed to a holistic understanding of the phenomenon of family resilience from parents with varied features [30], such as the different levels of their family resilience as well as general information. The Chinese version of Family Resilience Assessment Scale [31] was employed to measure family resilience among parents of adolescents with depression. The total score ranges from 44 to 176, with higher scores indicating a higher level of family resilience. After the first part of this project (a quantitative survey) was completed, we calculated the scores of participants’ family resilience on the minimum (81), 25% quartile (118), median (126), and 75% quartile (143) and the maximum (169), respectively. We selected parents with different family resilience scores from these four intervals as interviewees. Specifically, potential participants were asked to fill out the Family Resilience Assessment form, which took 5–10 min. The researcher calculated a family resilience score on the spot and decided whether to collect their qualitative data based on their family resilience score, as well as their age, employment, educational level, duration of care, etc.

Besides the above-mentioned sampling strategies, the inclusion criteria for the parents were: (1) caring for an adolescent (12–18 years old) diagnosed with major depressive disorder (International Classification of Disease, Tenth Version F32.2–32.3) for more than 12 months since the systematic treatment for depression generally lasts about 12 months which includes treatment at acute phase, maintenance phase and consolidation phase [32]; (2) their child experienced at least one recurrence to ensure that each participant had the experiences of family resilience at the different phases of illness; (3) able to comprehend and complete the questionnaires relevant to this study.

The first author, a female doctoral student with five years of clinical experience and experience in qualitative research, conducted all the interviews. Before collecting data, the first author worked at the research site for one year and had a certain understanding of adolescents with depression and their parents. Prior to the interviews, the researcher contacted the head nurse of the inpatient department of children and adolescent psychiatry, who was the person in charge of the department and had the most comprehensive understanding of all patients and caregivers in this department. Head nurse identified parents who were taking care of an adolescent with depression and met the inclusion criteria of this study as potential participants and asked their permission to consent to the researcher contacting them. If agreeable, the researcher then contacted them and provided written informed consent that described the purpose and procedures of the study (location and duration). They were guaranteed the right to withdraw at any time without negative consequences. They also were informed that all transcripts, the analysis of data, and presentation of results were anonymous. Then, all participants were asked to complete a self-report demographic questionnaire and the Chinese version of Family Resilience Assessment Scale to determine whether they met the criteria of the interviewees. The individual interviews were conducted by the first author in a psychological counseling room in the inpatient department. 

In-depth interviews were field audio-recorded and conducted in Mandarin (the required language of working environments in China). Data were collected in retrospect through face-to-face, semi-structured interviews, each lasting approximately one hour (range: 42–94 min). The parents were asked to talk as broadly as possible about their perceptions of family resilience at different phases of their child’s illness and responded to open-ended questions, including, “How did you and your spouse feel when your child was first diagnosed with depression?”; “What happened to your family life when your child was hospitalized?”; “What changes have taken place in your family after child’s discharge from the hospital compared with before the illness?”; “What’s family life like when your child relapses?”. Prior to recruitment, the study received ethical approval.

Despite data saturation origins in grounded theory, it is also applied in many other approaches to qualitative research, including IPA [33]. Data saturation occurred after the 11th interview; that is, no new content appeared [34]. In addition, we did three more interviews, which was defined as the number of additional interviews in which no new themes emerged [35]. Finally, this study involved 14 parents as participants. Detailed information about the 14 parents is presented in Table 1.

### 2.3. Data Analysis

Data analysis followed the process outlined for IPA by Smith et al. [29]. Firstly, the audio recordings from the in-depth interviews were transcribed verbatim by analyst 1. To ensure anonymity and confidentiality, the transcripts were numbered using “P1, P2…P14”. Two researchers (analyst 1 and analyst 2) repeatedly read the transcripts to engage more insight into each participant’s lived experience inductive thematic analysis occurred. Then, use more concise and conceptual phrases to annotate the transcripts in the right-hand margin with an inductive, iterative, and idiographic stance. Analyst 3 reviewed the audio recordings and transcripts to ensure the themes were representative and accurate. The research team (analyst 1, analyst 2, and analyst 3) had regular meetings to integrate the research findings and finalize the themes. 

## 3. Results

This study emerged with four higher-order themes with 12 sub-themes: (1) decompensation phase (newly diagnosed phase): (i) misinterpretations of illness, (ii) heavy psychological burden, (iii) chaotic rhythms in family; (2) launch phase (acute treatment phase): (i) potential influences of labeling, (ii) we must cure my child anyway, (iii) begin adjusting to family roles; (3) recovery phase (rehabilitation phase): (i) family reflection, (ii) subsequent reorganization of family resources, (iii) ultimately establishing a new balance; (4) normality phase (recurrence phase): (i) adaption for medical seeking process, (ii) actively lower expectations, (iii) concerns of future needs. In the following, the presentation of higher-order and sub-themes is expressed in explanatory quotes.

### 3.1. Decompensation Phase

Symptoms of depression in the early stage often made it difficult for parents to distinguish from their teenager’s rebellious behavior or even to believe that their child was faking it to avoid school or socialization. Most parents were usually skeptical of the diagnosis at the new diagnosis stage due to parents’ misconceptions about adolescent depression and inadequate explanation of the diagnosis by the treating physician. Once the misinterpretation is not solved, they might fall into a tremendous psychological burden. A chaotic rhythm in the family followed and made the family enter the decompensation phase.

#### 3.1.1. Misinterpretations of ILLNESS

Despite the doctor’s diagnosis, most parents were still skeptical about the diagnosis. Due to the disease misinterpretation, most parents exhibited some unreasonable behaviors during the newly diagnosed phase, such as refusing psychiatric medication, invoking the Buddha, and seeking psychological consultation excessively.

P7: *“…The doctor prescribed two kinds of drugs for her and gave her a diagnosis of depression. I thought that she (deliberately) behaved like depression and misled the doctor’s diagnosis, so I didn’t take drugs for her (daughter)…”*

P1: *“…The doctor said that my child had depression. I didn’t believe the diagnose…We even went everywhere to ask for Buddha’s help (the father attributed his son’s suicide behavior to being possessed by ghost). “*

P9: *“When she (daughter) was first diagnosed with depression, her father and I didn’t believe it…We took her to psychotherapy for more than a dozen times.”*

#### 3.1.2. Heavy Psychological Burden

After the disease misinterpretation was revealed, the parents began to fall into the heavy psychological burden and showed emotional grief, insomnia, frequent smoking, etc.

P9: *“…At that time, I really felt that the sky had fallen…I cried out of the hospital on the spot. I still want to cry up to now (getting wet with tears in the eyes).”*

P1: *“After confirming that he (son) had depression, I didn’t sleep all night. Sometimes I fell asleep for an hour or two at most…”*

P14: *“…At the beginning, I was very anxious. After my child was ill, I smoked more.”*

#### 3.1.3. Chaotic Rhythms in Family

The diagnosis of depression is undoubtedly a heavy blow to the family, and the original family rhythms are completely disrupted.

P2: *“…At that time, my family life was very chaotic. I didn’t want to cook, and I didn’t have the mind to care the family hygiene.”*

P13: *“When my daughter was just diagnosed, I immediately put down all my work to accompany her, and all my attention was on her. I couldn’t work and make money at that time.”*

### 3.2. Launch Phase

When the child was hospitalized at a psychiatric ward, the label of mental illness came with it. While bearing the shame of the mental illness, parents had great expectations and confidence in their children’s cure and recovery when they were hospitalized for professional treatment for the first time. Because of the confidence, the parents would actively use short-term strategies to deal with the sudden life changes brought about by hospitalization.

#### 3.2.1. Potential Influences of Labeling

With the gradual increase in illness cognition, these families increasingly felt that they were labeled with depression, resulting in social distance.

P2: *“This kind of illness (depression) was more or less linked to mental illness after all, so we couldn’t ask others for help. We could only solve the problem by ourselves.”*

#### 3.2.2. We must Cure My Child Anyway

Most parents had great confidence in the recovery and prognosis when the child was first hospitalized.

P8: *“I must have confidence in the child’s recovery. As long as I was alive, I would try to cure her.”*

#### 3.2.3. Begin Adjusting to Family Roles

With their children hospitalized for treatment, parents needed to respond quickly to sudden challenges, such as adjusting to multiple family roles.

P9: *“When I accompanied my daughter to the hospital, I begged my mother to come to my home and take care of my son (another child in the family) for a while.”*

### 3.3. Recovery Phase 

In the transition phase, the family gradually realized some challenges they needed to deal with and began to think about short-term coping strategies. During the recovery phase, the family started to consider the long-term coping patterns, including family reflection, integration of family resources, and building a new balance.

#### 3.3.1. Family Reflection

After a child was ill for a period of time, parents began to gradually accept the fact so that their sense of shame would be reduced, and they would actively think about the significance of adversity.

① *Reduced stigma.*

P9: *“When I brought my daughter to the hospital for the first time, I didn’t dare to use the medical insurance.I was afraid that there were records affecting her…Now I directly told others that it was “depression”…In today’s society, there were so many people with psychological diseases, and we also needed to ask others for help (so no longer hided the disease).”*

② *Significance of adversity.*

P4: *“I thought that this thing (the child was ill) was a hurdle for my family. If we could overcome this hurdle, it would be an inspiration for me and my child. We would be full of confidence in any difficulties in the future.”*

#### 3.3.2. Subsequent Reorganization of Family Resources

When the child came home from the hospital, one of the first things families needed to address was the issue of long-term care. The flexible family division of labor could help these families better deal with this problem. In addition, when the family was facing disease events, the family’s socio-economic resources could help the family recover better, including arranging their own internal resources reasonably and seeking various resources outside the family actively.

① *Flexible family division of labor.*

P3: *“I had been responsible for accompanying and caring her (daughter diagnosed with depression), and her mother was responsible for taking care of her brother (a two-year-old boy)…We were just temporary workers, so our family had a flexible time allocation.”*

② *Socio-economic resources.*

P14: *“I thought my family still had advantages over many families in dealing with child’ s illness. The first was my good economic foundation and the second was my well-connected resources…”*

#### 3.3.3. Ultimately Establishing a New Balance

The establishment of a new balance in the family was to let each family member actively participate in dealing with the events of the disease, learn to negotiate and solve problems, and improve the family atmosphere.

① *Family negotiation.*

P10: *“…I would pay more attention to the way I speak and be more active in caring for him (the sick son)…I discussed his future life plan with him and listened to his ideas carefully.”*

② *United family atmosphere.*

P5: *“My husband and I had both learned to understand each other, and our family was more united. When my husband came home (he worked in another city to earn money), he would help me do some housework. If he was tired, I would ask him to rest. He worked hard outside (to earn medical expenses), and I would understand him.”*

### 3.4. Normality Phase

Adolescent depression was a chronic disease, and recurrence was normal. When the child experienced a recurrence, the family recovery process would fluctuate to some extent, but it was more of a regular trend in the long run. The main manifestations of the family normality phase include: adapting to the medical process, actively lowering expectations, and concerns about future needs.

#### 3.4.1. Adaption for Medical Seeking Process

After many times of medical treatments, parents gradually accepted the reality that the child’s illness would recur, and they were able to reasonably arrange their family life when they were hospitalized again.

P10: *“…When we were first hospitalized, we were overwhelmed. On our second visit, I knew I would be in the hospital for more than ten days. I arranged everything at home and took him out (for hospitalization in another city)…”*

#### 3.4.2. Actively Lower Expectations

When a child’s illness was recurrent, especially after multiple hospitalizations, parents would actively lower expectations.

P3: *“I still expected her to recover, but with much less confidence…It (the illness) was been going on for a long time. As long as she was healthy, we didn’t expect much on her study…”*

#### 3.4.3. Concerns of Future Needs

Because of the instability of the illness, adolescents with depression needed to take a long leave of absence from school. Perhaps the important challenge for the family in the future would be school reintegration.

P10: *“The child had been out of school for a year, but his condition was still unstable… He tried to jump off a school building. The teacher found out, so we came back to the hospital again.”*

## 4. Discussion

This was a novel study exploring the dynamic processes of family resilience and its specific performance of each phase from the perspectives of 14 parents of adolescents with depression. The parents revealed that their families faced different challenges and adversity at different phases of their child’s illness (newly diagnosed phase, acute treatment phase, rehabilitation phase, recurrence phase). This might inform the continuing development of assessment, intervention, and policy strategies to reduce health disparities between families of adolescents with depression and ordinary adolescents in the future. Themes associated with the dynamic processes of family resilience in each phase identified in this research and its possible reasons and improvement strategies will be discussed below.

This study showed that many families had misinterpretations of illness and exhibited some unreasonable behaviors when their children were newly diagnosed with depression, which was consistent with the findings of a recent systematic review. Li and Reavley [36] found that the recognition rate of depression in Chinese mainland family caregivers was low, and they were more likely to regard psychosocial (such as stress) and personality (i.e., introversion) factors as important reasons for mental illness. Because the diagnostic approaches for adolescent depression were mainly based on patient interviews and subjective assessment of their clinical symptoms, it was difficult for parents to believe the diagnosis results from psychiatrists the first time, thus delaying the treatment of the disease [37]. Considering that the parents’ understanding of adolescent depression might affect their future help-seeking behaviors and parents often play a critical role in children’s treatment decision-making, mental health professionals should give timely explanations about the diagnosis of adolescent depression to the parents [38]. As Guidelines for Adolescent Depression in Primary Care (GLAD-PC) suggested, mental health professionals should not only take into account parents’ s cultural factors that might affect the diagnosis and management of this disorder but should also be aware of the negative reactions of parents to a possible diagnosis of adolescent depression (i.e., denial or sadness) [39].

In this study, family stigma was a prominent problem that needed to be noticed during the transition phase. This finding was similar to the results of Fitryasari et al. [27], that family stigma was an important reason affecting the family resilience among family caregivers of schizophrenic patients. In this study, family stigma was mainly discussed from the perspective of parents of adolescents with depression. Affiliate stigma refers to the internalization of public prejudice by family caregivers and directing it toward them, which includes the cognitive dimension (developing negative self-evaluation), affective dimension (having negative emotions), and behavioral dimension (concealing illness or refusing to seek help for others) [40]. Perlick et al. [41] have found that peer support was effective in reducing affiliate stigma, even better than health education on stigma by clinicians. For this, mental health professionals could organize peer-support meetings in adolescent psychiatric wards to promote families’ ability to cope with illness and its stigma. In addition, these parents should be encouraged to find their own language to describe their child’s condition (i.e., my child catches a cold of the soul) and to share the disclosure with their relatives and close friends to improve the gradual development of family resilience.

In the recovery phase, the family started to consider the long-term coping patterns, including family reflection, integration of family resources, and building a new balance. Noticeably, some families of adolescents with depression developed such an evident pattern of role specialization (i.e., the mother was responsible for more childcare, and the father was responsible for more paid employment) [42]. Although the family division of labor improved the establishment of a family organization to some extent, a long-term pattern of role specialization would make the mother or father who took on more childcare less well-being than their spouses [43]. Role specialization in and of itself was not problematic, but the division needed to be based on good family communication; that is, both sides agreed that this was a reasonable and fair division of labor and knew perspective-taking and negotiation. In addition to the reasonable division of labor and good family communication, the parents of adolescents with depression should look at problems from an advantageous perspective. For example, my child’s condition was relatively light and easy to recover; our family was relatively united and could withstand the disease crisis; our family had a good economic foundation and was able to cope with difficulties, etc.

Since the school was a primary discharge environment for adolescents with depression, school reintegration was an important challenge for families of adolescents with depression in the normality phase. A study found that some adolescents with experience of psychiatric hospitalization had considerable difficulty with school reintegration [44]. Compared with adolescents who had never been suspended, those who had returned to school following suspension encountered more adverse outcomes, such as low self-esteem, alcohol consumption, illicit drug use, high family conflict, low rank, and decreased satisfaction in their peer group [45]. In fact, prolonged school non-attendance in adolescents with depression has been a serious public health problem [46,47]. Fortunately, the Ministry of Education of the People’s Republic of China (2021) pointed out that the mental health institutions needed to further carry out the related public lecture on depression in middle schools and high schools, and each school needed to set up psychological consulting room and a mental health education curriculum, equipped with mental health education teachers [48]. However, a more individualized intervention approach under the comprehensive analysis of individual and environmental circumstances seemed to be more appropriate when dealing with the complex problem associated with school reintegration after psychiatric hospitalization [49]. Mental health professionals need to understand the real thoughts and actual situation about school reintegration among adolescents with depression and their parents and develop an appropriate discharge plan for them during the hospitalization.

### Limitations

This study was conducted at the inpatient department of children and adolescent psychiatry, which only allowed one caregiver to stay with the adolescent due to requirements for management in the psychiatric ward and prevention of the 2019 novel coronavirus. Hence, we can only invite one primary caregiver of adolescents with depression to participate in the interview. As each perspective is valued and valid in its own right, future research could also benefit from exploring the views and experiences of both parents of adolescents with depression in order to gain a fuller understanding of the impact of adolescent depression on family resilience. In addition, more dynamic information about the process of family resilience may be obtained through a longitudinal qualitative analysis, but this approach needs to take into account the change in the researcher’s experience and the loss of information caused by the loss of the interviewees and when explaining a certain phenomenon. Hence, this study used purposive sampling techniques to select the parents whose child’s disease duration was over 12 months and who experienced four phases (newly diagnosed phase, acute treatment phase, rehabilitation phase, and recurrence phase) to ensure a complete understanding of the family resilience process. Lastly, this study used a qualitative approach, and the results were qualitative in nature which need to be re-examined with a quantitative study design that can test the validity and reliability, which our research team had done in another survey. Despite our study’s limitations, our findings offer an interesting insight into the family resilience progress of adolescents with depression.

## 5. Conclusions

This study provided a better understanding of the dynamic process of family resilience from the perspective of the parents of adolescents with depression. In the decompensation phase, due to the misinterpretations of illness, parents had some unreasonable behaviors, which would delay the timely treatment of their child. In the transition phase, parents’ s affiliate stigma would isolate the family from the outside so as to make bad use of resources outside to help the family meet difficulties brought on by the illness. During the recovery period, families gradually thought about long-term coping strategies to help families establish a new balance. When the child experienced a recurrence, the family recovery process would fluctuate to some extent, but it was more of a regular trend in the long run. Mental health professionals could provide targeted suggestions to help parents of adolescents with depression achieve family resilience by assessing its different phases. Notably, this instrument must be validated before it can be used.

## Figures and Tables

**Table 1 ijerph-20-02564-t001:** General Information of Interviewed Parents.

Number	Relationship	Age (Years)	Employment	Educational Level	Marital Status	Number of Household Members	Number of Children	Duration of Care (Months)	Child’s Recurrence Times	Scores of Family Resilience
P1	Father	45	Employed	Junior high school	Married	4	2	15	2	131
P2	Mother	46	Unemployed	University	Married	3	1	18	2	155
P3	Father	45	Unemployed	Junior high school	Married	4	2	18	4	124
P4	Mother	39	Employed	Junior high school	Married	4	2	36	2	157
P5	Mother	46	Employed	Junior high school	Married	4	2	12	2	162
P6	Mother	47	Unemployed	Junior high school	Remarried	5	3	96	5	103
P7	Mother	40	Employed	High school	Widowed	3	2	16	2	142
P8	Father	43	Employed	High school	Married	4	2	36	5	148
P9	Mother	45	Unemployed	Junior high school	Married	4	2	48	4	127
P10	Mother	38	Unemployed	High school	Married	4	2	36	2	121
P11	Mother	47	Unemployed	Junior high school	Married	4	2	24	2	104
P12	Mother	52	Unemployed	Junior high school	Married	3	1	24	1	139
P13	Father	49	Employed	High school	Married	3	1	18	1	124
P14	Father	49	Employed	Master	Divorced	2	1	12	1	83

## Data Availability

Data available on request due to restrictions eg privacy or ethical.

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
