# Peer review of "Family Resilience Progress from the Perspective of Parents of Adolescents with Depression: An Interpretative Phenomenological Analysis"

_ijerph, 2023, doi:10.3390/ijerph20032564_

Round 1

Reviewer 1 Report

The paper submitted for review undoubtedly raises a very important problem, which is the depression of adolescents. What is important, the Authors try to identify family resources, so they present a salutogenic approach, not a pathogenic one. Family resilience is a construct that has been described in the literature relatively recently and therefore requires research. Research identifying resilient processes in families with specific problems of its members is important, e.g. somatic and mental health problems.

Taking up the topic and conducting research using the concept of family resilience is therefore a strength of the presented project. However, there are serious, in my opinion, weaknesses of the article:

1) Ambiguous anchoring of the study in the concept of family resilience. In the theoretical part, the authors only indicated what resilience is and referred to two models, including the Walsh model, which seems to have been the theoretical basis for the study. However, this is not unambiguous, because when presenting the empirical material and discussing the results of the study, the authors do not refer to either of these two models, i.e. they do not refer to the subject of the interviews to resilience processes.

2) Family resilience processes from the Walsh model have been described, but they are not reflected in the presented empirical material. Based on the theoretical introduction and the definition of the research goal, I would expect that the interviews would identify specific family resilience processes. However, this is not the case - citing fragments of interviews, the authors refer to the phases of the disease. Thus, the empirical material seems to refer more to the models of responding to loss and the phases (together with the corresponding empirical material) resemble the phases of coping with grief in the face of loss (in this case, loss of health).

3) The purpose of presenting the level of family resilience of the surveyed people (based on the FRAS questionnaire) also remains unclear. The authors presented the method of dividing the study group into quartiles and, therefore, one might expect that they would describe whether resilient processes were triggered and which of them in subjects with low, moderate, and high levels of family resilience. However, when presenting the empirical material, there is no reference to the level of family resilience. There is also no comment regarding this issue in the discussion.

4) In accordance with the title of the work, we could also expect that the content of the interviews describing the individual stages of coping with the disease by the adolescent's parents/family will show the dynamics of resilient processes, i.e., for example, the moments when these processes were launched and how they then developed will be identified, or how the family used resources in subsequent phases.

I am aware that the above remarks are general, but I hope they will be helpful in making the paper more consistent, including consistency of theory and empirical material. If the authors were satisfied with the above comments, the text would require major changes - redrafting of the text and a new interpretation of the content of the interviews.

Author Response

Response to Reviewer 1 Comments

Point 1: Ambiguous anchoring of the study in the concept of family resilience. In the theoretical part, the authors only indicated what resilience is and referred to two models, including the Walsh model, which seems to have been the theoretical basis for the study. However, this is not unambiguous, because when presenting the empirical material and discussing the results of the study, the authors do not refer to either of these two models, i.e. they do not refer to the subject of the interviews to resilience processes.

Response 1: Deeply appreciate for the comments. We deeply apologize that we have not clearly articulated the theoretical part in the introduction, which caused you misunderstanding. In fact, we did not adopt either of these two models as the theoretical basis for this study. This is because the first theory treats family resilience as a characteristic that does not explain well the family resilience process in parents of adolescents with depression. The second theory does not explore in depth the specific performance of family resilience processes in the different phases of illness. Considering that both previous theoretical models do not explain well the family resilience process of parents of adolescents with depression and its specific manifestations at different phases of illness, we conducted the present study. Our purpose in presenting these two theories is to illustrate our research significance in conducting this study, and we did not present them in our results and discussion because we did not consider them relevant to our research purpose. In addition, the interpretive phenomenological study requires a suspension of all suppositions, so we did not use previous theories as the theoretical basis for this study.

In the revised manuscript, based on your suggestion we restate the two previous theoretical models and the reasons why we did not choose them as the theoretical model. “Different authors have different understanding of family resilience. McCubbin, H. I. and McCubbin, M. A. [22] describe resilience as a family characteristic or pattern under stressful or adverse circumstances. However, the crisis faced by families is dynamic in nature and therefore; it is necessary to consider the dynamic process of family coping with illness when studying family resilience [23]. Walsh [16] emphasizes an ecological and developmental view in understanding family resilience and defines it as the function of family system in coping with adversity. Key progress in Walsh's Family Resilience Framework consists of three main components: (1)belief systems, including making meaning of adversity, positive outlook, transcendence and spirituality; (2)organizational patterns, including flexibility, connectedness, social and economic resources; (3)communication/ problem-solving, including clarity, open emotional expression, and collaborative problem-solving [16]. However, this framework does not explore in depth the specific performance of family resilience processes in the different phases of illness [24]. In order to facilitate a more comprehensive and nuanced perspective for exploring the family resilience process of parents of adolescents with depression and its specific manifestations at different phases of illness, this study divides the phase of illness of adolescent depression into four phases: newly diagnosed phase, acute treatment phase, rehabilitation phase, and recurrence phase. “Please find it on page 2, line 60-77.

Point 2: Family resilience processes from the Walsh model have been described, but they are not reflected in the presented empirical material. Based on the theoretical introduction and the definition of the research goal, I would expect that the interviews would identify specific family resilience processes. However, this is not the case - citing fragments of interviews, the authors refer to the phases of the disease. Thus, the empirical material seems to refer more to the models of responding to loss and the phases (together with the corresponding empirical material) resemble the phases of coping with grief in the face of loss (in this case, loss of health).

Response 2: We thank the reviewer for pointing out this issue. We did not adopt Walsh's Family Resilience Framework as the theoretical basis for this study. This is because the framework does not explore in depth the specific performance of family resilience processes in the different phases of illness. The purpose of this study is to explore the dynamic processes of family resilience in each illness phases among adolescents with depression from the perspectives of parents. Hence, this study was conducted to represent the family resilience process by describing the dynamics of family resilience at different stages of the illness through the perspective of parents of adolescents with depression. In the revised manuscript, based on your suggestion we restate Walsh's Family Resilience Framework and the reasons why we did not choose it as the theoretical model. Please find it on page 2, line 60-77.

Point 3: The purpose of presenting the level of family resilience of the surveyed people (based on the FRAS questionnaire) also remains unclear. The authors presented the method of dividing the study group into quartiles and, therefore, one might expect that they would describe whether resilient processes were triggered and which of them in subjects with low, moderate, and high levels of family resilience. However, when presenting the empirical material, there is no reference to the level of family resilience. There is also no comment regarding this issue in the discussion.

Response 3: Thank you very much for your instructive comments. Participants in this study were recruited by purposive sampling technique and maximum variance sampling strategy. We used both general demographic information and family resilience levels of participants as a sampling strategy to determine whether potential participants could be included in the interviews. In Table 1, we present the family resilience scores of different participants for the reader's reference. In addition, all materials in the results section are labeled with the participant's number (P1-P14), which can be combined with Table 1 to understand the level of family resilience scores. Since this study is a qualitative study rather than a quantitative study, the data on family resilience levels of the 14 participants may not reflect the overall family resilience levels of parents with adolescent depression, and therefore, after our careful consideration, the discussion section does not discuss the family resilience levels. Based on your suggestions, we have explained the purpose of presenting the level of family resilience of the surveyed people (based on the FRAS questionnaire) in the revised manuscript.

"Participants for this study were recruited by purposive sampling technique with maximum variation sampling strategy, which contributed to a holistic understanding of the phenomenon of family resilience from parents with varied features [30], such as the different levels of their family resilience as well as general information. The Chinese version of Family Resilience Assessment Scale [31] was employed to measure family resilience among parents of adolescents with depression. The total score ranges from 44 to 176, with higher scores indicating a higher level of family resilience. After the first part of this project (a quantitative survey) was completed, we calculated the scores of participates' family resilience on the minimum (81), 25% quartile (118), median (126), and 75% quartile (143) and the maximum (169), respectively. We selected parents with different family resilience scores from these four intervals as interviewees. Specifically, potential participants were asked to fill out the Family Resilience Assessment form, which took 5-10 minutes. The researcher calculated a family resilience score on the spot and decided whether to collect their qualitative data based on their family resilience score, as well as their age, employment, educational level, duration of care etc." Please find it on page 3, line 127.

Point 4: In accordance with the title of the work, we could also expect that the content of the interviews describing the individual stages of coping with the disease by the adolescent's parents/family will show the dynamics of resilient processes, i.e., for example, the moments when these processes were launched and how they then developed will be identified, or how the family used resources in subsequent phases.

Response 4: Thanks a lot for your kind suggestion for improving our work. In the results section, we have reinterpreted the content of the interviews according to your suggestions. “This study emerged four higher-order themes with 12 sub-themes: (1)decompensation phase (newly diagnosed phase): (i)misinterpretations of illness, (ii)heavy psychological burden, (iii)chaotic rhythms in family; (2)launch phase (acute treatment phase): (i)potential influences of labeling, (ii)we must cure my child anyway, (iii)begin adjusting to family roles; (3)recovery phase (rehabilitation phase): (i)family reflection, (ii)subsequent reorganization of family resources, (iii)ultimately establishing a new balance; (4)normality phase (recurrence phase): (i)adaption for medical seeking process, (ii)actively lower expectations, (iii)concerns of future needs. In the following, the presentation of higher-order and sub-themes is expressed in explanatory quotes.” Please find it on page 7, line 199.

Reviewer 2 Report

The manuscript Family resilience progress from the perspective of parents of adolescents with Depression: An interpretative phenomenological analysis by Zhang et al. presents a clinimetric approach to assess the strategies employed by families of depressed adolescent patients, providing an option for adolescent depression specialists. Despite the high quality of this manuscript, some points need to be refined to increase the quality of this manuscript. For example, the phrase Depression is a common mood disorder should be replaced by Depression affects 20% of the world's population. And at the end of the introduction, the authors should state that they are presenting the first version of an instrument with potential clinical utility. From line 148 onwards, the authors should change Author 1, 2, or 3 to analyst 1, 2, or 3. And the authors should state in their conclusions that this instrument must be validated before it can be used.

Author Response

Response to Reviewer 2 Comments

Point 1: The manuscript Family resilience progress from the perspective of parents of adolescents with Depression: An interpretative phenomenological analysis by Zhang et al. presents a clinimetric approach to assess the strategies employed by families of depressed adolescent patients, providing an option for adolescent depression specialists. Despite the high quality of this manuscript, some points need to be refined to increase the quality of this manuscript.

For example, the phrase Depression is a common mood disorder should be replaced by Depression affects 20% of the world's population. And at the end of the introduction, the authors should state that they are presenting the first version of an instrument with potential clinical utility. From line 148 onwards, the authors should change Author 1, 2, or 3 to analyst 1, 2, or 3. And the authors should state in their conclusions that this instrument must be validated before it can be used.

 Response 1: Thanks a lot for your kind suggestion for improving our work. The following is our point-by-point response to you.

1) We have replaced the phrase “Depression is a common mood disorder” to “Depression affects 20% of the world's population”. Please find it on page 1, line 28.

2) In the revised manuscript, we have stated that we are are presenting the first version of an instrument with potential clinical utility. “It is intended to lay the groundwork for future research to develop a theoretical framework or research instrument on family resilience in parents of adolescents with depression.” Please find it on page 2, line 95-97.

3) We have changed Author 1, 2, or 3 to analyst 1, 2, or 3. Please find it on page 8, line 188-196.

4) We have stated in the conclusions that this instrument must be validated before it can be used. Please find it on page14, line 454.

Round 2

Reviewer 1 Report

Thank you for the interesting text. I hope my comments have been helpful in improving the presentation of the paper.